# KEGG_Extractor: An Effective Extraction Tool for KEGG Orthologs

**DOI:** 10.3390/genes14020386

**Published:** 2023-02-01

**Authors:** Chao Zhang, Zhongwei Chen, Miming Zhang, Shulei Jia

**Affiliations:** 1Marine Sustainable Development Research Center, Third Institute of Oceanography, Xiamen 361102, China; 2Nantong Marine Environmental Monitoring Center, Ministry of Natural Resources, Nantong 226002, China; 3Third Institute of Oceanography, Ministry of Natural Resources, Xiamen 361005, China; 4Institute of Microbiology, Chinese Academy of Sciences, Beijing 100101, China

**Keywords:** KEGG, KEGG Orthology (KO), keyword matching, methodology

## Abstract

The KEGG Orthology (KO) database is a widely used molecular function reference database which can be used to conduct functional annotation of most microorganisms. At present, there are many KEGG tools based on the KO entries for annotating functional orthologs. However, determining how to efficiently extract and sort the annotation results of KEGG still hinders the subsequent genome analysis. There is a lack of effective measures used to quickly extract and classify the gene sequences and species information of the KEGG annotations. Here, we present a supporting tool: KEGG_Extractor for species-specific genes extraction and classification, which can output the results through an iterative keyword matching algorithm. It can not only extract and classify the amino acid sequences, but also the nucleotide sequences, and it has proved to be fast and efficient for microbial analysis. Analysis of the ancient Wood Ljungdahl (WL) pathway through the KEGG_Extractor reveals that ~226 archaeal strains contained the WL pathway-related genes. Most of them were *Methanococcus maripaludis*, *Methanosarcina mazei* and members of the *Methanobacterium*, *Thermococcus* and *Methanosarcina* genus. Using the KEGG_Extractor, the ARWL database was constructed, which had a high accuracy and complement. This tool helps to link genes with the KEGG pathway and promote the reconstruction of molecular networks. Availability and implementation: KEGG_Extractor is freely available from the GitHub.

## 1. Introduction

Genome sequencing has penetrated many fields of bioscience [1]. It also has the potential to completely change medical care [2,3]. There is an increasing need for more bioinformatics methods to make full use of the genome sequencing data. As an important step, automatic gene functional annotation is essential in interpreting genomic data. The Kyoto Encyclopedia of Genes and Genomes (KEGG, http://www.genome.jp/kegg/, accessed on 24 September 2022) is a widely used reference database used for understanding the advanced functions and the components of biological systems [4,5,6,7,8]. Currently, KEGG contains 18 databases, which are divided into three categories (system information, genome information and chemical information) [9]. As one of the core databases of KEGG, the KEGG Orthology (KO) database—a large, manually curated collection of protein families—serves as a baseline reference to link genes with pathways through K number identifiers [10]. In the KEGG Orthology database, genes that perform the same function are grouped together; these are known as the ortholog groups (KO entries). Each KO contains multiple pieces of gene information and plays a role in one or more paths. Generally, KOs represent a class of gene clusters with similar functions, and they are defined in the context of the KEGG pathway and other molecular networks [10]. Therefore, the whole KEGG pathway and molecular networks can be automatically reconstructed if the genes are given the K numbers. At present, about 12,934,525 (48%) protein sequences are assigned to KOs in the KEGG database [11]. The repertoire of KO entries is being expanded through continuous efforts, and automated interpretation of high-level functions are conducted through the improved collection of KEGG modules. Genome annotation in KEGG involves complicated computational procedures and can be manually checked for further analysis [12,13,14]. Briefly, members of a KO group are selected, defining a core subgraph in the SSDB graph, which is then computationally extended by the KOALA (KEGG Orthology and Links Annotation) program [15]. The existing KEGG tools, such as the GhostKOALA, BlastKOALA [16] and the KEGG Automatic Annotation Server (KAAS, http://www.genome.jp/tools/kaas/, accessed on 24 September 2022) [17], are available to assign protein sequences with KOs. They use the BLAST+ [18] and GHOSTX [19] software to search amino acid sequences against the GENES database. The KO assignment of the GENES database is performed by the previously developed KOALA tool and the newly developed KoAnn tool. Another newly developed KEGG annotation tool—KofamKOALA—is a web server that is widely used to assign KEGG Orthologs to protein sequences via a homology search against a database of profile Hidden Markov Models (pHMMs) with adaptive score thresholds [20]. The HMM database with an adaptive score threshold is named KOfam (HMM database customized by KEGG Orthologs). KofamKOALA [20] is based on KOfam and assigns KO numbers to the submitted sequences using the HMMER/HMMSEARCH software. Sequence matching with a score higher than the threshold is considered more reliable than other matching, so it is highlighted with ‘*’ in the output of these tools [20]. Thus, the results with asterisks can be considered the best hits and true positives.

K number assignments have promoted the association analysis by linking the user’s sequence data to the KEGG pathways and EC numbers, which have greatly improved the efficiency of gene annotation [20]. The performance of the available KEGG tools has also proved that the KofamKOALA is faster than any other KO distribution tools, and its accuracy is comparable with the best tools [20]. In addition, the KofamKOALA has additional functions to send the results to KEGG mapper automatically for reconstruction of pathways (PATHWAY), hierarchical function classifications (BRITE) and pathway modules (MODULE) [20]. It has wide applications in gene annotations of comparative genomics and metagenome [21,22]. However, although the available KofamKOALA and other KEGG tools can ensure large numbers of reliable annotations, the current tools cannot quickly extract and classify all the corresponding gene sequences (amino acids/nucleotides) from different species. For example, the KofamKOALA can only output statistical information without detailed sequences and species classification information. At present, publicly available sequence extraction tools mainly involve the Seqkit [23], the Seqtk, the Bedtools [24] and the Gffread [25]. They can extract gene sequences from the genomes according to the location of different genes, but they can neither recognize the KEGG annotation results nor extract the amino acid sequences. Another sequence extraction tool, TBtools [26], has general functions of sequence extraction, but it cannot be operated on batch genomes. The extracted results with TBtools also lack species annotation information, which is a shortcoming of this tool. We still lack an effective tool for species-level gene extraction of the KEGG results, which hinders the subsequent microbial genome analysis. Here, we proposed an effective tool, KEGG_Extractor, to solve the defect of the KEGG tool. It is developed based on a pattern detection algorithm and can quickly extract and classify the sequences of the KEGG annotations of KofamKOALA (the widely used KEGG tool). It stores all the extracted sequences according to different KO assignments and adds the species information to the results, showing the species-specific gene sequences. In this article, we showed the usage, steps and applications of the tool, and compared the recall rate, accuracy assessment and operation efficiency of the KEGG_Extractor based on different KEGG software. The following protocol completely describes how to use KEGG_Extractor to extract and classify the annotated results of KEGG from the related genomes in detail.

## 2. Materials and Methods

### 2.1. KEGG Orthology Search Tools

(1) The KOfam database (https://www.genome.jp/ftp/db/kofam/, accessed on 24 September 2022) and the KofamScan programme were downloaded and installed. After decompressing the KOfam database, the path of the profiles directory and the ko_list file were checked to ensure the correct location of the database. The detailed installation methods referred to the official documents for guidance (https://ftp.genome.jp/pub/tools/kofam_scan/INSTALL, accessed on 24 September 2022).

(2) The dependent software for KofamKOALA had also been downloaded and installed, including the Ruby (version ≥ 2.4) and HMMER (version ≥ 3.1). The running command of KofamKOALA was exec_annotation and the E-value parameter was set to 1 × 10^−5^. After homologous searching in the genomes, the high score assignments highlighted with asterisks ‘*’ in the output were chosen for further analysis.

### 2.2. Gene Filter and Construction of the Non-Redundant Gene Dataset

Seqkit is a cross-platform and ultrafast toolkit for FASTA/Q file manipulation, which can meet common requirements. It can be installed directly through conda (conda install seqkit). The “seqkit --help” command can be used to gain more information about a command.

The CD-HIT is software used to cluster protein or nucleic acid sequences. It can quickly cluster sequences according to their similarity to remove redundant sequences. Thus, the latest CD-HIT v4.8.1 software [27] was chosen and installed to remove the redundant genes. The parameter was set as follows: -c 1 -n 5 m 1600 -d 0 -T 8.

### 2.3. Python3 and the Genomic Annotation Files

The Python programming language (Python v3.7 and the above versions) was downloaded and installed; at the same time, it was added to the directory search path environment. Then, a set of input genomic annotation files ending with faa (protein) and fna (cds) was prepared, from which the genes could be extracted in the next steps.

### 2.4. The Assembly Summary File

The folder named species containing the assembly summary files (interval with tab) had also been prepared ahead of time. It was worth mentioning that the species information must be laid in columns 8 and 9 of the summary files. For example, the reference archaeal assembly file with the standard input format was downloaded from NCBI (https://ftp.ncbi.nlm.nih.gov/genomes/refseq/archaea/assembly_summary.txt, accessed on 5 January 2022).

The assembly summary file contained the complete species information of each archaeal genome, which could be linked to the corresponding extracted genes by the KEGG_Extractor.

### 2.5. The Whole Genomic Annotation Sequences

The reference archaeal genomes (protein and cds files) were downloaded from the NCBI Assembly (https://ftp.ncbi.nlm.nih.gov/genomes/refseq/archaea, accessed on 5 January 2022).

### 2.6. Accuracy Assessment of the Constructed Database

To evaluate the accuracy of the extracted genes by the KEGG_Extractor, an artificial dataset including 54 non-redundant cdhA genes (acetyl-CoA decarbonylase/synthase complex subunit α) was constructed from the NCBI and the UniProt databases to evaluate the accuracy of the constructed archaeal WL-pathway database. The artificial dataset was searched against the constructed WL-pathway database (ARWL), KEGG, COG and the SwissProt databases using diamond at 40–100% identity to compare their accuracy. The matched and well-annotated sequences were considered true positives, while sequences with unpredicted gene names were considered false negatives.

## 3. Results and Discussion

### 3.1. Implementation and Performance Assessing

The KO database is a collection of orthologs, most of which are members of the KEGG molecular networks, including KEGG pathways, BRITE hierarchies and KEGG modules. Using the genome annotation procedure in KEGG, about 46% of 17 million genes can be annotated with K numbers [16]. Therefore, the sequence similarity search against the GENES database is a search for the most appropriate K numbers, which can easily be computerized as implemented in the KOALA programs. Thus, after achieving sequence alignments and KEGG annotation of all the proteins using the KofamKOALA, as mentioned above, the generated results are merged and processed for the next sequence extraction and classification. Unless specified otherwise, the program will use default values for the parameters. The workflow for KEGG_Extractor is shown in Figure 1A. The tool uses the logic of keyword matching and extracts the gene sequences from the genome files by reading the information of the specific columns of the input files. If the option -i is specified, the corresponding coding sequences of all genes will be scanned based on the annotation files generated by the KEGG tools as the program runs. Option -f will automatically read the processed files of the KEGG annotations with KO entries, while option -s means that the species information of each gene will be checked from the genomic assembly files and added into the results. Examples can be checked and downloaded from the online data of GitHub (https://github.com/xielisos567/KEGG_extractor, accessed on 24 September 2022. Command-line use of the KEGG_Extractor is shown below: python3 KEGG_extractor.py [-options]/{-i <directory> -f <processed files> -s <assembly files> -o <directory>} (Figure 1B). 

According to the characteristics of keyword matching, we have embedded iterative rules based on keyword matching algorithm in the KEGG_Extractor, which is efficient for data analysis. Significantly, this tool can not only extract species-level amino acid sequences; it can also correspondingly extract nucleotide sequences and automatically output the extracted sequences according to different KO assignments. Finally, the gene sequences will be classified and organized by category according to the given K numbers. The extracted nucleotide sequences are conductive to the abundance analysis in the metagenome, while the extracted amino acid sequences are conducive to the construction of gene dataset and sequence alignment.

To compare the performance of the KEGG_Extractor based on different KEGG tools such as the KofamScan, BlastKOALA and KAAS, we used twenty reference archaeal genomes randomly selected from the NCBI database as test queries. Firstly, two HMM profile-based KOs (K00192 and K00195) of the Wood–Ljungdahl pathway (M00377) were chosen to search the genomes using the KEGG tools, as stated above. The results with the most hits were further chosen and filtered. Then, through the KEGG_Extractor, the merged results were read by specified column and extracted accordingly from the twenty reference archaeal genomes. Finally, we obtained the resulting folder with the given KO numbers. Each of the KO folders contained twenty species-level gene sequence files. The exported genes were laid, respectively, according to their genome sources (Appendix A). The test set contained forty sequences with KOs corresponding to the KOfam database. All the extracted gene sequences have been further checked in the NR (Non-Redundant Protein) database of NCBI (https://www.ncbi.nlm.nih.gov, accessed on 5 January 2022), and the results show no false positive results. For the 20 genomes constituting our test set, the performance was comparable among the KEGG_Extractor with KofamScan (recall: 1), BlastKOALA (recall: 0.925) and GhostKOALA (recall: 0.975), while the KEGG_Extractor with KAAS showed a lower recall (0.85). In total, 40 genes were extracted through the KEGG_Extractor based on the KOfamScan, 37 based on the BlastKOALA, 39 based on the GhostKOALA and 34 based on the KAAS (Table 1). Obviously, the KofamScan performs best among the KEGG software. In terms of the CPU time, KofamScan was 69, 2.1 and 1.1 times faster than BlastKOALA, KAAS and GhostKOALA, respectively [20]. KofamScan can benefit more from the reduction of the target database compared to the other three tools, while it is among the tools showing the highest *F*-measures [20]. Thus, the performance of the KEGG_Extractor based on the KofamScan (KofamKOALA) was better than any other KEGG-based tool used for sequence analysis. Users can also customize KOfam by choosing a subset of KOs so that they can focus on the annotation of specific class of proteins while reducing the computational time. In this case, the runtime of the KEGG_Extractor based on the KofamKOALA had also been tested. Regarding the CPU time, the real time of the KEGG_Extractor is 1.281 s for amino acid sequences and 1.626 s for the nucleotide sequences, which is fast for the gene extraction. Currently, no tools have the function of sequence extraction for microbiological analysis, so this tool proved to be fast and accurate for gene analysis based on the KEGG annotation results.

### 3.2. Step-By-Step Example of the Usage of KEGG_Extractor

Microorganisms play an important role in bio-geochemical cycling, which is important in carbon cycling. Methanogenesis coupled with the Wood–Ljungdahl (WL) pathway/reductive acetyl-CoA pathway is the largest carbon fixation pathway under anaerobic conditions and is one of the most ancient metabolisms for energy generation and carbon fixation in the Archaea [28,29,30]. Carbon monoxide dehydrogenase/acetyl-CoA synthase (CODH/ACS) is a five-subunit enzyme complex responsible for the carbonyl branch of the WL pathway (Table 2). The fact that the four CODH/ACS subunits are homologous between archaea (CdhABCDE) and bacteria (AcsABCDE) indicates that this enzyme existed before the divergence between the two life domains [31], so it is of great significance to study the diversity of this gene clusters (Table 2). We carried out an exhaustive search for homologs of all six subunits in 1199 reference genomes of the Archaea (5 January 2022). Taking the archaeal protein sequences as an example, the procedure was as follows:

Step 1: Firstly, the KO annotation was conducted using the KofamKOALA. All the target HMM profile-based KOs were placed in the profiles folder and the ko_list profile was also downloaded ahead of time. Then, after the annotation of all the genomes, the results highlighted with asterisks ‘*’ were selected and merged for further analysis (Figure 2). An example of the detailed code was shown in Figure 2, while batch genomes could be executed with the loops. It is worth mentioning that the E-value of 1e-5 performs well, avoiding false-positive results, and it is the default value in many sequence alignments. The merged result file could be implemented as follows: 

for i in `ls ./|grep out$`; do sed -i ‘1, 2d’ ./$i && less ./$i | awk -v var=“$i” ‘{print var”\t”$0}’ >> archara_WL_results_all.out; done

grep “*” archara_WL_results_all.out > merged_KofamKOALA_result.txt

Step 2: The generated and merged results (e.g., merged_KofamKOALA_result.txt) of step 1 should be prepared as the input file of the next step. Meanwhile, the folder named species1 containing the summary file of the reference archaeal genomes, as stated in the Materials and Methods section, was another input file. All the archaeal protein sequences (faa format) were placed in the archaea_protein folder, from which the final gene sequences would be scanned and correspondingly extracted. Then, the code could be conducted as stated in Figure 3. The KEGG_extractor.py script would read the merged file of the KofamKOALA and scan the archaeal protein sequences to match the information according to the first and second column of the merged file. Once the connection had been established, it would extract the gene sequences and place them correspondingly into the created KOs folders according to the third column of the merged file. Meanwhile, the KEGG_extractor.py script could scan in the species1 folder according to the matched first column to match its corresponding species information and add the species information to the results. Eventually, it would output the results containing the target KOs folders, and each of the KO folders contained the gene sequences with species information (Figure 3). For example, in the WL pathway of all the archaea, a total of seven KOs folders were generated in the result, and each of them contained a different number of gene sequence files (Figure 3).

Lastly, since there are many genes with partial sequences in the NCBI database (not well sequenced), the extracted genes need to be further filtered out to obtain complete reference gene sequences for subsequent analysis. Thus, we filtered the extracted genes according to the standard gene length. Such processes can be executed with the help of dependent tools such as the Seqkit (https://github.com/shenwei356/seqkit, accessed on 24 September 2022), which is a special software used to process fasta/fastq sequence files. The gene length statistics could be conducted with the following executable code: seqkit fx2tab --length –name --header-line extracted_seqs.faa/extracted_seqs.fna. For example, the reference length of the CO dehydrogenase/acetyl-CoA synthase complex subunit α (K00192) was ~ 806 aa, so we filtered the sequences far less than 806 aa, such as the partial sequence length of ~ 400 aa. Then, the filtered genes were eventually merged with the merged file for the next step of the analysis.

Step 3: All the filtered gene sequences were combined to form a combined file (e.g., WL_all.faa). Then, the combined file conducted a de-redundancy on the CD-HIT software (100%) and the non-redundant gene dataset was generated for further analysis. The executable code was shown as follows: cd-hit -c 1 -n 5 m 1600 -d 0 -T 8 -i WL_all.faa -o NonRundant_WL_prot (Figure 3). The extraction of nucleotide sequences followed the same process as mentioned above. The program would extract the corresponding gene sequences from the annotated nucleotide files by identifying the protein accession numbers. Then, the extracted sequences would be merged, undergoing a de-redundancy implemented by the CD-HIT (100%).

### 3.3. Applications of the KEGG_Extractor in Genomic Analysis

CODH/ACS has been generally thought to be a feature of most methanogenic archaea (Figure 4A), as well as some acetogenic, carboxydotropic, aceticlastic and sulfate-reducing bacteria [31,32]. After genome annotation with the KofamKOALA in 1199 reference genomes of the Archaea (5 January 2022) and sequence extraction with the KEGG_Extractor, the extracted sequences underwent de-redundancy using CD-HIT (identity: 100%). A total of 982 non-redundant genes of archaea were finally obtained, and 226 strains of the archaea contained genes related to the WL pathway. Most of them were *M. maripaludis*, *M. mazei* and members of the *Methanobacterium*, *Thermococcus* and *Methanosarcina* genus. It is reported that the CODH/ACS complexes existed at least in the genomes of *Methanoregula boonei* [33], *Methanoregula formicica* [34], *Methanocella arvoryzae* [35] and one member of Methanomicrobiales (*Methanospirillum hungatei*). Our results are consistent with the available studies and have further added many potential new species (Figure 4B). Once again, this confirms the reliability of the KEGG_Extractor tool in gene extraction and classifications.

As stated above, the genes extracted by the KEGG_Extractor provided a reference archaeal dataset of the WL pathway (ARWL). The partial genes or the not satisfiable genes were filtered through the Seqkit. Furthermore, we also compared the accuracy of the newly constructed ARWL database with that of other databases. The results showed that all methane cycling genes of the WL pathway were correctly assigned to their targeted gene families at the 40% and 50% sequence similarity in the four databases. When the sequence similarity increased to 60%, there existed two false negative genes (sequences with their gene names not predicted) in the four databases (Figure 5). When the threshold improved from 70% to 100%, the false negative rate of the newly constructed database ARWL increased from 13% to 35%. In contrast, the false negative rate of KEGG database increased from 17% to 65%, that of the COG database increased from 19% to 80%, while that of the SwissProt database increased from 20% to 81% (Figure 5). In particular, when the sequence similarity reached 100%, the number of sequences without predicted gene names was 43 and 44 in the COG and the SwissProt database, respectively. However, there were only 19 false negative genes in the WLDB database, indicating a high species-specific characteristic of this database. Therefore, the results showed that the ARWL database performs better over a 70% identity threshold than the other KEGG tools. It represented high accuracy and comprehensiveness over other orthology databases in methane cycling analysis of the WL pathway. The ARWL database has several important features for analyzing CH_4_ cycling processes compared to the currently available orthology databases, such as the COG, SwissProt and KEGG. It contained the most comprehensive gene families in the ancient WL pathway of methane cycling. These gene families are based on the most current knowledge and up-to-date research progress in methane cycling of the archaea. This also indicated that the ARWL database constructed through the KEGG_Extractor performed well in the sequence accuracy, and it had high species specificity.

## 4. Conclusions

Currently, the KEGG_Extractor can deal with most microbial genomes annotated by the available KEGG tools, and it achieved fast and accurate genomic analysis based on the KofamKOALA software. This tool can be used not only for database construction but also for species statistics; thus, it is undoubtedly a practical tool for microbial genome analysis. The constructed database was proved to be more accurate and complete than the other databases. Overall, we believe that in this work, we provide the community with an effective tool in conjunction with the current KEGG tools to profile microbiomes for species-level genes, as well as a comprehensive process of gene extraction, filtering and database construction.

## Figures and Tables

**Figure 1 genes-14-00386-f001:**
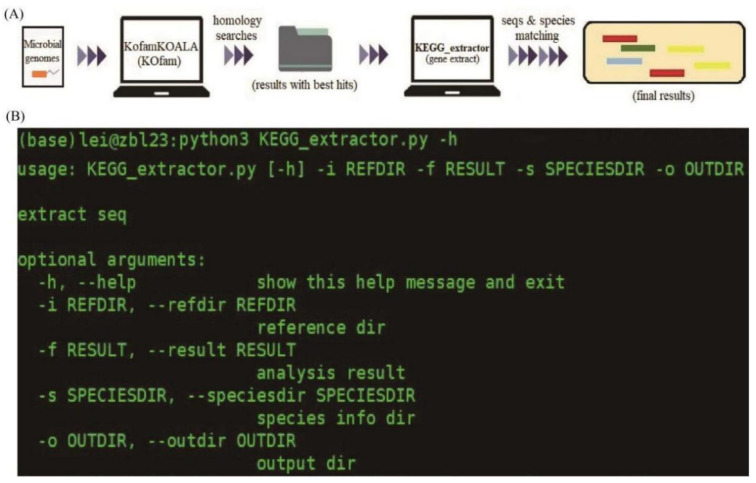
Workflow (**A**) and parameter (**B**) of the KEGG_Extractor tool.

**Figure 2 genes-14-00386-f002:**
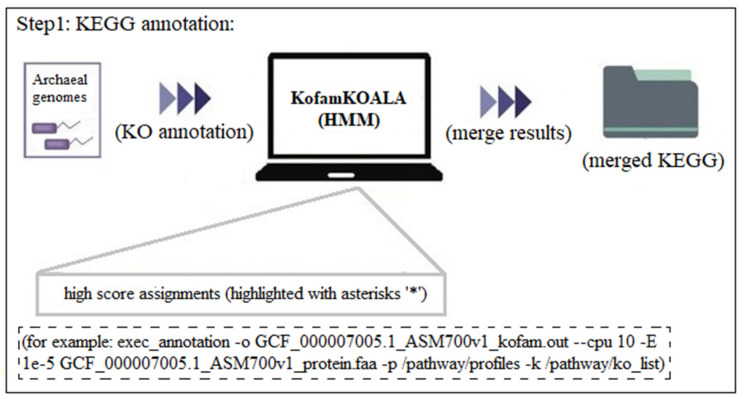
Step 1 of the usage and applications: KEGG annotation.

**Figure 3 genes-14-00386-f003:**
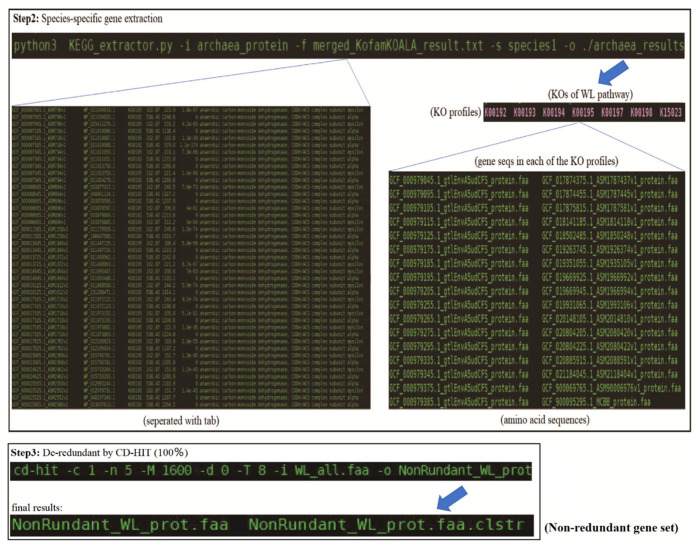
Step 2 and step 3 of the usage and applications: species-level gene extraction and de-redundancy through CD-HIT (100%).

**Figure 4 genes-14-00386-f004:**
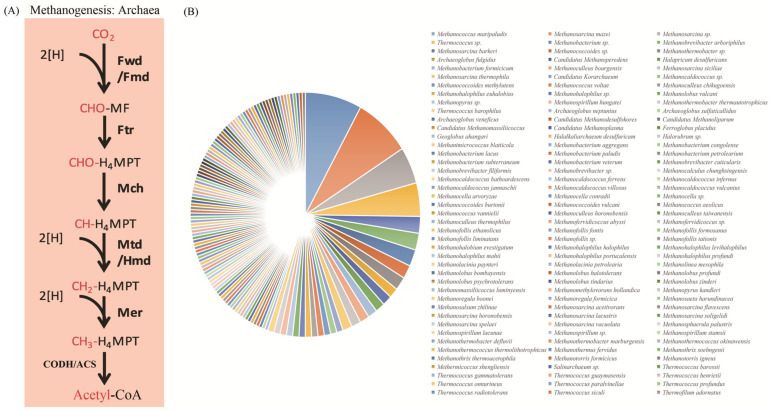
(**A**) The reactions of WL pathway in the archaea. The final step of this reaction is achieved through the CODH/ACS enzymatic complex. (**B**) Predicted strains of the archaea containing the WL pathway.

**Figure 5 genes-14-00386-f005:**
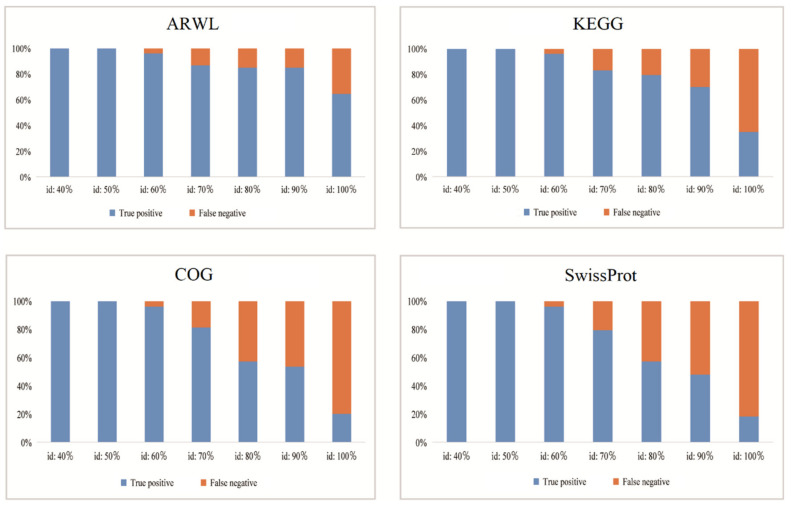
Accuracy assessment of the ARWL, KEGG, COG and SwissProt databases.

**Table 1 genes-14-00386-t001:** Performance of the KEGG_Extractor based on different KEGG tools.

Types	KOfamScan	BlastKOALA	GhostKOALA	KAAS
Number of extracted genes	40	37	39	34
True positive rate	100%	100%	100%	100%
Recall	1	0.925	0.975	0.85

**Table 2 genes-14-00386-t002:** Wood–Ljungdahl pathway of the Bacteria and Archaea.

Pathway (Entry)	KOs	Key Enzymes	Substrate	Strains
Wood–Ljungdahl pathway (M00377)	K00198	cooS, acsA; anaerobic carbon monoxide dehydrogenase catalytic subunit [EC:1.2.7.4]	CO_2_	Bacteria (Firmicutes, Planctomycetes, Deltaproteobacteria, Spirochaeata); Archaea (Euryarchaeota)
	K15023	acsE; 5-methyltetrahydrofolate corrinoid/iron sulfur protein methyltransferase [EC:2.1.1.258]		
	K14138	acsB; acetyl-CoA synthase [EC:2.3.1.169]		
	K00197	cdhE, acsC; acetyl-CoA decarbonylase/synthase, CODH/ACS complex subunit γ [EC:2.1.1.245]		
	K00192-K00195	cdhABCD; anaerobic carbon monoxide dehydrogenase, CODH/ACS complex subunit		

## Data Availability

The source code for the KEGG_Extractor can be found at GitHub (https://github.com/xielisos567/KEGG_extractor, accessed on 24 September 2022).

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
