# Peer review of "KEGG_Extractor: An Effective Extraction Tool for KEGG Orthologs"

_genes, 2023, doi:10.3390/genes14020386_

Round 1

Reviewer 1 Report

Dear Author/

Thank you for this valuable paper in this recent field. Needing major revision, English editing, material reconstruction, presenting results in a simple form.

Reviewer 2 Report

The authors have presented an interesting work. KEGG_extractor tool will be beneficial to the scientific community to profile microbiomes for species-specific genes and functional analysis.

I have a few suggestions  for the authors to improve the MS:

Please rephrase the materials and methods in past tense so that it doesn't read like an instruction manual. For eg. Lino no. 52 can be rephrased as "Python programming language was downloaded".

Line no. 91 can be rephrased as "Microorganisms play a crucial role in bio-geochemical cycling, importantly in carbon cycling (reference)".

Reviewer 3 Report

The proposed software would be extremely useful for practice and the paper is important in this respect. However, the quality of the manuscript must be improved before publication. 

The major thing that I find missing from the paper is a thorough step-by-step example of the usage and application of the proposed software, which would then be helpful for readers to understand the usage of the software so that they can use it in their own research. Unless this can (potentially) happens, the publication in such a top journal would not be meaningful. So, the author must include such details, with screenshots, codes and resulting figures/tables, for the usage of their proposed software for practical applications (such illustration is currently also not available from the GitHub page of the software that is referred to in the paper). 

I could not find any supplementary material also it is referred to at the end of the paper. Authors must check this too. 

Finally more details about the KEGG and its contains, with illustrations, would be useful to make the paper understandable for a wider audience group (who may not be expert in KEGG but wish to perform such gene related research) increasing the usefulness of the proposed software and hence that of the paper. 

Round 2

Reviewer 1 Report

Dear Author/ 

Thank you for your efforts in rewriting the paper in an accepted form.

Reviewer 3 Report

I have no further comments.